# Digital Apps to Improve Mobility in Adults with Neurological Conditions: A Health App-Focused Systematic Review

**DOI:** 10.3390/healthcare12090929

**Published:** 2024-04-30

**Authors:** Reem Rendell, Marina Pinheiro, Belinda Wang, Fiona McKay, Ashleigh Ewen, Catherine Carnegie, Erin Tikomaidelana, Zino Fattah, Leanne Hassett

**Affiliations:** 1Sydney School of Health Sciences, Faculty of Medicine and Health, The University of Sydney, Sydney, NSW 2000, Australia; 2School of Health Sciences, Western Sydney University, Sydney, NSW 2000, Australia; 3Ingham Institute for Applied Medical Research/South Western Sydney Local Health District, Sydney, NSW 2000, Australia; 4Institute for Musculoskeletal Health, The University of Sydney/Sydney Local Health District, Sydney, NSW 2000, Australia; 5Sydney School of Public Health, Faculty of Medicine and Health, The University of Sydney, Sydney, NSW 2000, Australia; 6School of Health and Social Development, Faculty of Health, Deakin University, Melbourne, VIC 3000, Australia; 7Institute for Health Transformation, Faculty of Health, Deakin University, Melbourne, VIC 3000, Australia; 8Royal Rehab, Sydney, NSW 2000, Australia; 9Illawarra Shoalhaven Local Health District, Wollongong, NSW 2500, Australia; 10South Western Sydney Local Health District, Sydney, NSW 2000, Australia

**Keywords:** neurological, rehabilitation, mobile app, mHealth, eHealth, systematic review, MARS, ABACUS

## Abstract

The provision of mobility exercises through a smartphone application (app) for people undertaking neurological rehabilitation may improve mobility outcomes. However, it is difficult for clinicians and consumers to select high-quality, appropriate apps. This review aimed to identify (1) which mobile health (mHealth) apps are suitable for prescribing mobility exercises for adults with neurological health conditions, (2) how well these apps incorporate telehealth strategies, and (3) how well these apps rate in terms of quality and capacity for behaviour change. The Australian Apple iTunes Store was systematically searched, by using a search code and manually, for apps suitable for training mobility in neurological rehabilitation. Additional searches were conducted in known app repositories and for web-based apps. Trained reviewers extracted data from the included apps, including population-specific characteristics; quality, by using the Mobile App Rating Scale (MARS); and behaviour change potential, by using the App Behaviour Change Scale (ABACUS). The included apps (n = 18) provided <50 to >10,000 exercises, many incurred a subscription fee (n = 13), and half included telehealth features. App quality was moderate (mean MARS score of 3.2/5 and SD of 0.5), and potential for behaviour change was poor (mean ABACUS score of 5.7/21 and SD of 2.1). A limited number of high-quality apps are available for the prescription of mobility exercises in people with neurological conditions.

## 1. Introduction

With an ageing population and the increasing prevalence of non-communicable and chronic diseases, a growing number of people are seeking to access rehabilitation globally, increasing demand and putting greater pressure on services delivering therapy programs [1]. Guidelines recommend that rehabilitation services provide adults undertaking neurological rehabilitation with large amounts of scheduled therapy and facilitate active task practice outside of therapy sessions [2,3,4]. Several strategies can facilitate additional practice, with no one strategy being identified as superior [5]. Despite this, at present, rehabilitation services struggle to meet these activity guidelines [6,7,8].

Digital interventions, such as virtual reality video games, activity monitors, and handheld computer technologies, are among the identified strategies to facilitate active task practice in and outside of scheduled therapy [5]. In particular, the use of technology to deliver rehabilitation remotely (termed telerehabilitation) [9], was identified during the COVID-19 pandemic as a potential solution to increasing the reach of rehabilitation by addressing distance and access problems [10]. Interviews conducted with participants using a range of technologies for mobility exercises in addition to their usual rehabilitation found that people were able to engage with technology both in the hospital and remotely at home, when prescription was tailored by a therapist, with optimal engagement relying on the right level of support [11].

Mobile health applications (apps) may constitute an affordable and scalable digital health intervention to address growing rehabilitation needs. Mobile device use is widespread worldwide, even accessible in low-resource settings, offering unique opportunities for service delivery [10]. A 2017 report identified that smartphone penetration rates are over 80% globally, including evidence of heavily populated middle-income countries such as China, India, and Brazil exhibiting over 80% smartphone ownership [12]. In an Australian context, 92% of Australians own a smartphone and 58% monitor health or fitness attributes on a smart device (smartphone, smartwatch, fitness band, etc.) [13]. 

Emerging evidence suggests that physiotherapy interventions delivered through a smartphone or tablet app could improve mobility outcomes in neurological populations [14,15,16]. Furthermore, people undertaking neurological rehabilitation report enjoying the use of technology, including apps, as part of their rehabilitation [17]. Thus, the use of mobile health (mHealth) apps is likely to be a feasible, acceptable, and effective means of accessing and delivering rehabilitation. However, with over 350,000 mobile health and fitness apps available for download worldwide in the Apple iTunes and Google Play stores and little regulatory control over their content [18], it is difficult for both clinicians and people undertaking rehabilitation to select high-quality, appropriate, and safe apps. Specifically, apps to support neurological rehabilitation need to cater to people with non-motor impairment (e.g., cognitive, sensory, and behavioural impairment) in addition to physical impairment, be appropriate for use via telehealth, and include behaviour change strategies to support sustained use for ongoing physical activity. Therefore, there is a need to identify and evaluate mHealth apps suitable for neurological rehabilitation. This research addresses this gap by systematically searching and reviewing apps to answer the following questions:Which mHealth apps are suitable for prescribing mobility exercises in adults with neurological health conditions?How well do these mHealth apps incorporate telehealth strategies?How well do these mHealth apps rate in terms of quality and capacity for behaviour change?

## 2. Materials and Methods

### 2.1. Design

This systematic review follows standard recommendations for traditional systematic reviews outlined in the Preferred Reporting Items for Systematic Reviews and Meta-Analyses (PRISMA) guidelines [19] and published conduct and reporting recommendations for systematic app store reviews [20]. The review protocol was prospectively registered on Open Science Framework (https://doi.org/10.17605/OSF.IO/35P7M). 

### 2.2. Search Strategy

As this study was conducted in Australia, the Australian Apple iTunes store was searched. A technology consultant established a search code to systematically search the iTunes Store by using an Application Programming Interface. Apps had to be available for download in Australia and contain a predefined keyword in either their title or description. Search terms included population (e.g., neurological) OR intervention (e.g., task-specific training) keywords (see Appendix A). Following an initial pilot search, the final search code was run on 17 June 2022. In a verification process of the automatic search code, a manual screen of the iTunes store with key search terms was completed in July 2022. Similarly to a systematic review grey literature search, we searched two known rehabilitation app repositories, i.e., MyTherappy (https://www.my-therappy.co.uk/ (accessed on 18 July, 2022)) and Mobile Health App Database (https://mhad.science/en/ (accessed on 18 July 2022)), and included known web-based rehabilitation apps for eligibility screening. Due to the complexity of updating the automatic search, the included apps were reviewed again in the iTunes store in December 2023 to show currency.

### 2.3. App Selection Criteria

Apps from the iTunes store were included for screening if they were available in English; updated within 18 months of the search date (cut-off date: 17 December 2020); and listed their primary genre as Lifestyle, Health, and Fitness or Medical. Apps requiring a companion device, product, or software, were included if the research team was able to easily access the companion device or software for testing, noting that the app was tested and not the companion device or software itself. Paid, subscription-based, and free apps were included. If an app had a paid (or “pro”) version and a free (or “lite”) version available, the paid version of the app was included. 

#### 2.3.1. Population

Apps were included if they targeted or would be suitable for adults and/or older adults undertaking neurological rehabilitation. This included health conditions such as stroke, traumatic brain injury, Parkinson’s disease, multiple sclerosis, and spinal cord injury.

#### 2.3.2. Intervention 

Apps were included if they permitted prescription of repetitive task-specific exercises targeting mobility limitations that a physiotherapist or similar health professional working with an adult neurological population is likely to prescribe. Exercises could include whole task practice, e.g., walking; part-practice where whole task components are practiced, e.g., stepping; or balance exercises, e.g., tandem standing. Apps providing exercises typically not prescribed for neurological conditions (e.g., back strengthening) were excluded. 

### 2.4. Data Management and Selection Procedure

The app store search output was downloaded into an excel spreadsheet. A three-step process was used to screen apps for inclusion. First, a single reviewer (R.R.) excluded apps not in English, that had not been updated within 18 months, or that did not list their primary genre as Lifestyle, Health, and Fitness or Medical. Next, two reviewers (R.R. and B.W.) independently screened apps for inclusion by using app store title and description (resembling traditional title and abstract screening). Screening outcomes were compared, and discrepancies were resolved via consensus or a third reviewer (L.H.). Apps potentially meeting the criteria were downloaded onto an Apple iPhone. If an app required a security step, log in, or had limited features when downloaded, the reviewer accessed the “developer website” or “app support” function on the app store to assess the app. Some apps were excluded at this time based on website content if it was clear the inclusion criteria were not met. When apps offered only a demonstration for new users or had limited access without consultation, the developer was emailed for a meeting with the lead reviewer (R.R.) and given two weeks to respond. 

The third step involved one reviewer (R.R.) reviewing apps on an iPhone for inclusion. A second reviewer, B.W., screened a random sample of 20% of apps to compare reviewer agreement. There was excellent agreement between reviewers (94%); therefore, one reviewer screened the remaining apps. This is a variation from the initial protocol, which was modified due to pragmatic reasons, including time and funding, and informed by excellent agreement between reviewers.

### 2.5. Data Extraction 

A team of independent reviewers who were health professionals with clinical experience in neurological rehabilitation conducted the app review. The included apps were downloaded onto the reviewers’ mobile phones. The reviewers used an iPhone 7 or newer and used the latest compatible iOS software for their phone (iOS 15 or 16). Data extraction was completed for each app independently by two reviewers by using a customised Qualtrics survey (Qualtrics, Provo, UT, USA, license held by The University of Sydney; see Appendix A). Data extraction included app information, technical aspects, theoretical background/strategies, telehealth features, target population, suitability for moderate-to-severe impairment (accessibility features), and types of exercises provided. Data were compared, and discrepancies were resolved via discussion. If consensus could not be reached, a third reviewer (B.W. or L.H.) was consulted.

The quality and behaviour change potential of included apps were evaluated by using two established rating scales (Mobile App Rating Scale [21] and App Behaviour Change Scale [22]) with data extracted from the same Qualtrics survey. Both scales enable customisation for the population being evaluated. Scale customisation was discussed and reviewed by a panel of neurological physiotherapists with clinical and research expertise.

#### 2.5.1. Mobile App Rating Scale

App quality was rated by using the Mobile App Rating Scale (MARS) [21], which has been validated and has good-to-excellent reliability and high objectivity [23,24]. It consists of 23 items, each containing a 5-point scale, from 1 (inadequate) to 5 (excellent). Apps are rated across four dimensions: Engagement, Functionality, Aesthetics, and Information. The MARS was customised to reflect the population group as suggested by Stoyanov et al. [21]. Mean scores were calculated for each dimension and the total, rating app overall quality. The MARS includes two additional optional dimensions; we included only the subjective quality dimension, as the other dimension (behaviour change) was covered by our other rating scale. If a rating score for an item differed between reviewers, the average of the two reviewer scores was calculated. If one reviewer rated an item “not applicable” in the Information dimension whilst the other scored the item, this was resolved through discussion between the two reviewers. 

#### 2.5.2. App Behaviour Change Scale

The app potential to change the user’s behaviour was rated by using the App Behaviour Change Scale (ABACUS) [22]. This consists of 21 items grouped into four categories: Knowledge and Information, Goals and Planning, Feedback and Monitoring, and Actions. Apps are given one point when an item on the scale is met and can achieve an overall score between 0 and 21 points, where a higher score indicates higher potential for behaviour change through app use. Any discrepancies in ratings were resolved via discussion or a third reviewer (B.W.). 

### 2.6. App Reviewer Training

The lead investigator (R.R.) developed a two-hour online training package detailing the app data extraction and rating process, consisting of five online training modules and a freely available online MARS training video (https://www.youtube.com/watch?v=25vBwJQIOcE (accessed on 8 September 2022)). As part of the training process, each reviewer practiced extracting data and rating the same two apps that had been previously excluded. Ratings between reviewers were compared, and any discrepancy, defined as greater than two points for each MARS question or any difference in answers for ABACUS questions, was reviewed and further training and rating practice provided. 

### 2.7. Analysis

Data on app description, target population, and exercise type were tabulated and descriptively synthesised. MARS and ABACUS subscale and total scale scores were tabulated for each app and ranked highest to lowest based on each scale’s total score. 

Given the high percentage of Australians owning Android phones, particularly in older age groups [25], the Android version of the top five rated apps (based on combined MARS and ABACUS rankings) available on iTunes were searched for and downloaded from Google Play in December 2023. Android apps were compared to their Apple counterparts in a verification process. This step was undertaken to confirm whether the top iTunes apps identified were also available for Android phone users and to compare their quality and behaviour change potential.

## 3. Results

### 3.1. Flow of App Review Process

The automatic iTunes search identified 29,263 apps excluding duplicates (Figure 1). A total of 1480 additional apps were identified through the manual app store search, web-based search, and through review of known app repositories. After titles and app store description screening, 155 apps were downloaded and screened for eligibility (see Appendix A for full list). Finally, 137 apps were excluded, leaving 18 apps included in the review. When the apps were checked again in December 2023 in the iTunes store, three were no longer available, and four had not been updated (Table 1).

### 3.2. Characteristics of Included Apps

Of the eighteen included apps, two were web-based apps and sixteen were downloaded through the iTunes store. Only six of the sixteen apps in the app store had published online reviews (38%) on a five-point star rating system, with most having between one and seven reviews. Physiapp was an outlier, with 422 reviews. Ratings across included apps ranged from 1 to 5 stars (with a higher number indicating a higher app rating) with a median of 3.9 stars. Of the eighteen apps, five (28%) were free, and thirteen (72%) required an ongoing subscription. One app required a companion device to function (LusioMATE). Fourteen (78%) apps required account set up, and thirteen (72%) required internet access to function (Table 1).

Nine (50%) out of the eighteen apps had an educational aspect in addition to exercises, providing information on a specific disease type/condition or relevant health promotion topics, e.g., falls prevention. Four (22%) apps included a means of completing an assessment or uploading assessment results, e.g., uploading six-minute walk test distance. The ability to customise exercises ranged from complete ability to customise all aspects (n = 7, 38.9%) to the provision of only pre-set programs with no ability to customise exercises (n = 5, 27.8%) (Table 1).

### 3.3. Suitability for Mobility Training in Neurological Rehabilitation

Apps were categorised as targeting a specific rehabilitation population based on type of exercises included in the app, the information, or images provided (Table 2). Six (33%) apps either solely targeted a specific neurological health condition (e.g., Parkinson’s disease) or included exercises for specific health conditions. The remaining apps targeted general rehabilitation, where whole-body functional exercises were available (n = 12, 67%); neurological rehabilitation, where exercises targeting very weak muscles were included (n = 6, 33%); and older adults (n = 7, 39%), where images, information, exercise aims, and affiliations clearly reflected the needs of older individuals. Only one app included exercises that would be suitable for all the different rehabilitation cohorts (PhysioTherapy eXercises (PTX)).

Apps were also evaluated on their accessibility features, specifically, design suitability or capacity for modification to accommodate people with moderate-to-severe impairment common among neurological health conditions (Table 2). Impairment accommodated for included physical (e.g., large buttons on display) (n = 5, 28%), language (e.g., voice-over screen reader option) (n = 15, 83%), perceptual/visual (e.g., larger text or magnification options) (n = 5, 28%), cognitive (e.g., simple design) (n = 12, 67%), and behavioural (e.g., reminders during exercises to maintain focus) (n = 5, 28%) impairment. Only one app was suitable for all impairment types (LusioMATE).

Table 3 provides data on the exercises included in the apps. The number of exercises provided varied considerably, ranging from providing less than fifty exercises to more than ten thousand exercises. Exercises provided different targeted mobility tasks, with eight apps (44%) including exercises for all mobility tasks evaluated. Additional exercises beyond mobility exercises (e.g., fitness training and wheelchair skills) were also included in fifteen (83%) apps. Twelve apps (67%) included exercises that were suitable for people with moderate-to-severe physical impairment. One app (LusioMATE) was a gamified app, so had no images of exercises. For the remaining seventeen apps, all had images (photos or drawings) or video demonstrations of the exercises. However, only five included images or videos of people of an appropriate age for the target cohort, e.g., an older person for stroke rehabilitation. One app (PhysioTherapy eXercises (PTX)) specifically included people from the target health condition in all exercise depictions.

### 3.4. Telehealth Features

Limited telehealth features were available in many of the included apps. Nine (50%) apps allowed for the remote access of the prescriber to view user data and remotely alter the program, but only one of these asked the user to consent to data sharing with the prescriber (PhysioTherapy eXercises (PTX)). One app also allowed the provider to send written feedback via a chat function (PhysiApp). Twelve (67%) apps allowed the user to record or track their progress (Table 1).

### 3.5. Quality and Capacity for Behaviour Change

The MARS app quality mean score was 3.2 (SD of 0.5) (Table 4). Most apps rated poorly on interest, customisation, and interactivity, reflected in the overall mean Engagement subsection score of 2.6 (SD of 0.6). Apps generally rated well based on Functionality (mean of 3.6 and SD of 0.7). Only three apps (n = 3, 17%) received a score for item nineteen, evidence base, indicating they have been trialled and reported on in the scientific literature.

ABACUS total scores were poor overall, with a mean total score of 5.7 (SD of 2.1) on a 0-21 scale (Table 5). The Knowledge and Information subscale saw the highest mean score, 2.6/5 (SD of 0.9). Only one app achieved a score for the Goals and Planning subsection (LusioMATE). The Feedback and Monitoring subscale was very poorly rated, with a mean of 1.7/7 (SD of 1.5), as was the Actions subscale, with a mean of 1.3/6 (SD of 0.6).

### 3.6. Apple–Android Comparison

A comparison of the top five ranked apps available on iTunes with the Android versions showed no or only minor differences in features and interface which would not have changed their overall ratings for quality and behaviour change potential.

## 4. Discussion

By using a systematic identification approach, this review identified a limited number (n = 18) of smartphone apps available in Australia with the potential to improve mobility in adults with a neurological health condition. This is the first review of its kind in this population group and intervention that we are aware of and the first time additional criteria pertinent to this population group have been explored in apps. The identified apps were generally of low-to-moderate quality, rarely incorporated behaviour change strategies into their design, and had limited telehealth capabilities. When considering pragmatic app elements necessary for use in a clinical setting, such as accessibility features, appropriateness of images, and exercise number and customisability, many apps fell short. Current and future apps could be improved by enhancing their capacity for population-specific exercise prescription, lowering costs to the prescriber, enhancing their evidence base, and boosting focus on behaviour change capabilities. Overall, the findings from this study have the potential to improve clinicians’ capability to select suitable apps for neurological rehabilitation and impact app developers’ decisions about what features to include in apps for this population group by providing detailed criteria for evaluating apps for this population.

The ratings on the MARS and ABACUS demonstrated widespread gaps in app engagement, evidence base and goal setting capability. The MARS results demonstrate that apps were poorly rated on their user engagement, including fun, interest, customisation, interactivity, and appropriateness for the target audience. These factors are critical to sustaining a user’s interest over time [26]. There is also a clear lack of evidence supporting individual app use. This is likely reflective of the rapidly changing nature of smartphone apps due to the commercialisation of app stores and the absence of regulation on app development, harnessing a perceived ease of app creation without the need for testing prior to consumer use. ABACUS ratings identified that only one app included a goal setting and review feature (LuisoMATE). Goal setting is widely used in rehabilitation and has been shown to harness the motivation required for behaviour change [27] and improve health-related quality of life and self-efficacy [28]. App developers should consider adding goal setting as a key feature when developing future apps.

The pragmatics of app use in a clinical setting are not considered in MARS or ABACUS scores, e.g., features such as exercise number and customisability, telehealth features, accessibility features, and cost to the prescriber. For an app to be useful to the prescriber, it should have enough exercises to design an appropriate program, with progression, addressing different mobility limitations. The apps in this review ranged from having less than fifty to over ten thousand exercises available for prescription, with less than half including exercises for all mobility tasks. The number of exercises available in each app was not necessarily indicative of the quality or relevance of these exercises to the population group, as the research team did not always consider these exercises reflective of routinely prescribed exercises for task-specific mobility training. There is likely a fine balance between apps that do not have sufficient exercises available for adequate exercise prescription and apps that have a very large number of exercises available, which may become overwhelming for novice clinicians who may not be able to effectively navigate and select the appropriate exercises.

Only one app (PhysioTherapy Exercises (PTX)) included age- and condition-appropriate images of people performing exercise, although qualitative data suggest that adults undergoing rehabilitation place importance on the age appropriateness of feedback-based technology [29]. Many apps provided opportunities for the user to record their own behaviour and for the prescriber to access and alter the program remotely, but only one app required the user to consent to this data sharing (PhysioTherapy Exercises (PTX)). Accessibility features (e.g., larger text) improve the ability for people with a neurological health condition to use smartphone applications [30,31], but despite most apps in this study including an accessibility feature, only one app included features for all impairment domains (LusioMATE). Due to the commercialisation of app development, most apps in this review incur a subscription cost to the prescriber, likely creating another barrier to widespread use of apps for exercise prescription.

Similar reviews of smartphone apps have been published in other population groups, including using the MARS to rate app quality for low-back pain [32], total hip and knee replacement surgery [33], cardiac rehabilitation [34], asthma [35], and pain [36]. In this sample of reviews, the included app numbers ranged from 9 to 218 (median of 23 apps and IQR of 15 to 61 apps) and mean MARS scores ranging from 2.4 to 3.4 (mean of 3.06 and SD of 0.41). This highlights that our review identified a similar number of suitable apps (n = 18) and that apps for mobility training in neurological populations are of a similar quality compared to other populations (MARS mean of 3.2 and SD of 0.49). Studies have also investigated the clinical effectiveness of apps in different population groups, such as in diabetes mellitus [37] and cardiovascular disease [38], with favourable results. The growing use of mobile technologies suggests that digital health and mHealth apps are likely to form a large part of the healthcare landscape in the future, although further research and app design need to be conducted to strengthen their quality and demonstrate effectiveness to support their use.

The strengths of this review include the methodology, specificity of data extraction, and reviewer experience. Reviewers were all health professionals with clinical experience in rehabilitation and underwent extensive training for data extraction. The methodology used follows standard recommendations for traditional systematic reviews outlined in the PRISMA guidelines [19] and follows recommendations for conducting and reporting app store systematic reviews [20]. This included prospective protocol registration, a systematic search strategy, screening and data extraction by independent reviewers, and quality assessment using published scales. We modified the MARS and added additional criteria to reflect the needs of adults undertaking neurological rehabilitation by consulting a panel of experts. The use of the ABACUS adds another dimension to app ratings not captured by the MARS. Through behaviour change modalities, it is conceivable that an app has further potential to improve mobility in adults undergoing rehabilitation. We believe that the modified MARS and the inclusion of the ABACUS provide a useful tool for clinicians for assessing the quality, usability, telehealth capabilities, and behaviour change potential of apps in the future, as well as serving as a guide for app developers when designing apps for this population group.

Limitations exist due to the continuous and abundant nature of apps and their rapid development. The large volume of available apps identified through the automatic search code (n = 29,263) led to challenges in limiting the records identified; thus, a pragmatic approach was employed to reduce the search output through strict search terms and exclusion criteria, risking missing relevant apps. We addressed this through an additional manual app store search and review of known app repositories and web-based apps. Some apps known to the research team that provide suitable exercises for adults with neurological health conditions were excluded due to their last update date. Furthermore, three of the eighteen apps included in the study are no longer available for download in the Australian iTunes Store, highlighting the challenge of identifying and using mHealth apps in practice. This indicates that a limitation of smartphone apps is the perpetual app development process and the need for regular updates to keep up with operating systems and ensure future usability. Moreover, due to the commercialisation of apps, we cannot be certain that the included apps comply with software development standards, which would enable them to be scaled, tested, and maintained over time. 

In our protocol, we had planned to present a combined ranked score for quality and behaviour change but decided against this, presenting the ranking of each scale separately. We made this decision as we felt that the combined score did not add to the identification of the best app, particularly due to the importance of additional features (e.g., cost, number, and type of exercises) when selecting the most appropriate app for a particular patient or context. Additionally, these scales do not evaluate all technical aspects of apps, e.g., battery and security; however, we included additional technical aspects that we felt are relevant for considering the suitability of apps for clinical practice, such as the requirement of a login/account and providing consent for data sharing. Finally, this review includes only the perspective of the clinician. Ideally, these apps should also be reviewed by people with neurological health conditions and by app developers, both of whom may place emphasis on different app aspects. Future research could investigate patient consumer and developer preferences for important attributes of apps suitable for neurological rehabilitation.

## 5. Conclusions

This review did not identify any clear recommendation for a single specific app for use in the clinical setting but identified a limited number for clinicians to choose from. The use of the MARS and ABACUS provided a ranking of apps based on quality and behaviour change capacity. In addition, we provided information on the pragmatics of each included app, including tech requirements, telehealth features, accessibility features, and type and quantity of exercises provided. This information will enable clinicians to look at a short list of apps and consider which features are the most important and suitable for their use. This information will also be useful for app designers, in the development and modification of apps to better meet the needs of the population.

## Figures and Tables

**Figure 1 healthcare-12-00929-f001:**
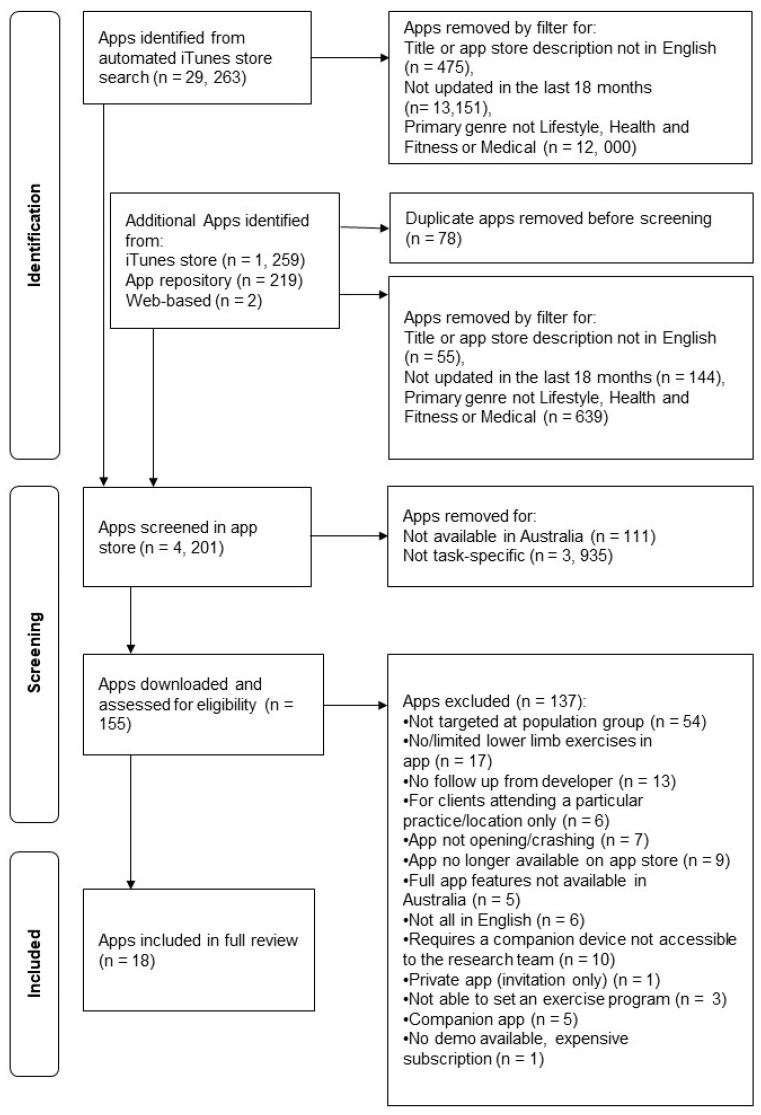
PRISMA flow diagram.

**Table 1 healthcare-12-00929-t001:** App description. Shaded apps were no longer available for download in Australia in December 2023.

Name (Version), Affiliation	App Store Star Rating (Number of Reviews)	Cost	Tech Aspects		Theoretical Background/Strategies			Telehealth Features			
			Account Required (Y/N)	Internet Access Required (Y/N)	Education Aspect (Y/N)	Assessment Aspect (Y/N)	Customisation *	Remote Prescriber Access to View Patient Data: Y/N; If Yes: ^#^	Remote Prescriber Access to Provide Feedback (Y/N)	Remote Prescriber Access to Alter Program (Y/N)	User Recording Their Own Behaviour (Y/N)
110 Fitness (V1.4) ^θ^, commercial	NR (0)	Subscription	Y	Y	Y	N	A	N	N	N	N
Connected mHealth (V1.3.0), commercial	NR (0)	Subscription	Y	Y	N	N	D	Y (B)	N	Y	Y
Daily Dose PD (V7.800.1), commercial	4 (1)	Subscription	Y	N	Y	N	B	N	N	N	N
LusioMATE (V1.4.3) ^©^, commercial	5 (2)	Subscription	Y	N	N	N	C ^@^	Y (B)	N	Y	Y
My Exercise Program (V1.0.4) ^θ^, commercial	NR (0)	Subscription	Y	Y	N	Y	D	Y (B)	N	Y	N
My TRcare—Stroke Exercises (V1.1.2), commercial	NR (0)	Free	Y	Y	N	Y	A	N	N	N	Y
PhysiApp (V2.2.2), commercial	3.7 (422)	Subscription	Y	N	Y	Y	D	Y (B)	Y	Y	Y
Physiotec (V1.8.4), commercial	1.9 (7)	Subscription	Y	Y	N	N	D	Y (B)	N	Y	Y
PhysioTherapy eXercises (PTX) (V W2.0.0), University	NA (web-app)	Free	N ^α^	Y	N	N	D	Y (C)	N	Y	Y
PhysioTools Trainer (V1.0.904) ^θ^, commercial	1 (1)	Subscription	Y	Y	N	N	D	Y (B)	N	Y	Y
Rehab Guru Client (V3.0.3), commercial	NR (0)	Subscription	Y	Y	N	N	D	Y (B)	N	Y	Y
Rephysio (V2.1.5) ^θ^, unknown	NR (0)	Subscription	Y	Y	Y	N	A	N	N	N	Y
Swiss Parkinsons (V1.6.1), NGO	NR (0)	Free	N	Y	Y	Y	C	N	N	N	Y
Track Rehab (N/A), commercial	NA (web-app)	Subscription	N	Y	N	N	D	Y (B)	N	Y	Y
Yoga Vista (V3.6.5), unknown	NR (0)	Subscription	Y	N	Y	N	C	N	N	N	N
Cleo—My MS App (V1.11.7), commercial	5 (1)	Free	Y	N	Y	N	A	N	N	N	Y
Get Steady—Balance Exercises (V1), commercial	NR (0)	Free	N	Y	Y	N	B	N	N	N	N
PhysioEd. (V6), commercial	NR (0)	Subscription	Y	Y	Y	N	A	N	N	N	N

Key: ^θ^ App was not updated between initial review date and December 2023. ^©^ App requires a companion device to function; NR: not rated; NA: not applicable. ^α^ Requires an account for some features. * A: Pre-set program (no customisation); B: pre-set program with modifiable dose OR difficulty; C: customisable exercises without dose OR difficulty customisation; D: completely customisable (exercise type, difficulty, AND dose). ^#^ A: With data sharing; B: without data sharing; C: without data sharing following consent. ^@^ Therapist can use sensors to design exercises but cannot modify dose adequately—1 min or long play only.

**Table 2 healthcare-12-00929-t002:** App target population suitability. Shaded apps were no longer available for download in Australia in December 2023.

App Name and Version	App Target Population				App Suitability for Moderate to Severe Impairment (Accessibility Features) ^&^				
	General Rehabilitation	Neurological Rehabilitation	Specific Health Condition ^#^	Older Adults	Physical	Language	Perceptual/Visual	Cognitive	Behavioural
110 Fitness (V1.4)	✕	✕	PD	✕	✕	✕	✕	✕	✕
Connected mHealth (V1.3.0)	✓	✕	✕	✕	✕	✓	✕	✓	✕
Daily Dose PD (V7.800.1)	✕	✕	PD	✕	✓	✓	✕	✕	✕
LusioMATE (V1.4.3)	✓	✓	✕	✕	✓	✓	✓	✓	✓
My Exercise Program (V1.0.4)	✓	✕	✕	✕	✕	✓	✕	✓	✕
My TRcare—Stroke Exercises (V1.1.2)	✕	✓	✕	✕	✕	✓	✕	✓	✕
PhysiApp (V2.2.2)	✓	✓	✕	✓	✓	✓	✕	✓	✓
Physiotec (V1.8.4)	✓	✓	✕	✕	✕	✓	✕	✓	✕
PhysioTherapy eXercises (PTX) (V W2.0.0)	✓	✓	Stroke, TBI, SCI (various levels), PD, and MS	✓	✕	✓	✓	✓	✕
PhysioTools Trainer (V1.0.904)	✓	✓	✕	✕	✓	✕	✕	✓	✓
Rehab Guru Client (V3.0.3)	✓	✕	✕	✕	✓	✓	✕	✓	✓
Rephysio (V2.1.5)	✓	✕	Stoke, PD, and MS	✓	✕	✓	✓	✓	✕
Swiss Parkinsons (V1.6.1)	✕	✕	PD	✕	✕	✓	✕	✓	✓
Track Rehab (v3.03.3)	✓	✕	✕	✓	✕	✓	✕	✓	✕
Yoga Vista (V3.6.5)	✓	✕	✕	✓	✕	✓	✓	✕	✕
Cleo—My MS App (V1.11.7)	✕	✕	MS	✕	✕	✓	✓	✕	✕
Get Steady—Balance Exercises (V1)	✕	✕	✕	✓	✕	✓	✕	✕	✕
PhysioEd. (V6)	✓	✕	✕	✓	✕	✕	✕	✕	✕

Key: ^#^ PD: Parkinson’s disease; MS: multiple sclerosis; TBI: traumatic brain injury; SCI: spinal cord injury. ^&^ See Appendix A for how these were rated.

**Table 3 healthcare-12-00929-t003:** Exercises. Shaded apps were no longer available for download in Australia in December 2023.

		Exercise Focus									Images of Exercises	
App Name and Version	Number of Exercises in App ^	Sitting	Standing Balance	Standing Up	Walking	Stair Climbing	Running	Strength Training	Additional Exercises	Exercise Suitability for People with a Moderate–Severe Physical Impairment (Y/N)	Images (Drawings and/or Photos and/or Videos) of Exercises (Y/N)	Patients in Drawings and/or Photos and/or Videos (Y/N/Some)
110 Fitness (V1.4)	B ^¤^	✕	✓	✕	✕	✕	✕	✓	✓	N	Y	Some
Connected mHealth (V1.3.0)	D	✓	✓	✓	✓	✓	✓	✓	✕	Y	Y	Some
Daily Dose PD (V7.800.1)	C ^¤^	✓	✓	✓	✓	✕	✓	✓	✓	Y	Y	N
LusioMATE (V1.4.3)	A ^∞^	✓	✓	✓	✓	✓	✓	✓	✓	Y	N	NA
My Exercise Program (V1.0.4)	D	✕	✓	✓	✓	✓	✓	✓	✓	N	Y	N
My TRcare—Stroke Exercises (V1.1.2)	C	✓	✓	✓	✓	✕	✕	✓	✓	Y	Y	N
PhysiApp (V2.2.2)	F	✓	✓	✓	✓	✓	✓	✓	✓	Y	Y	N
Physiotec (V1.8.4)	F	✓	✓	✓	✓	✓	✓	✓	✓	Y	Y	N
PhysioTherapy eXercises (PTX) (V W2.0.0)	D	✓	✓	✓	✓	✓	✓	✓	✓	Y	Y	Y
PhysioTools Trainer (V1.0.904)	G	✓	✓	✓	✓	✓	✓	✓	✓	Y	Y	N
Rehab Guru Client (V3.0.3)	G	✓	✓	✓	✓	✓	✓	✓	✓	N	Y	Some
Rephysio (V2.1.5)	A ^β^	✓	✓	✓	✓	✕	✕	✓	✓	Y	Y	N
Swiss Parkinsons (V1.6.1)	B	✓	✓	✓	✓	✕	✕	✓	✓	Y	Y	N
Track Rehab (v3.03.3)	E	✓	✓	✓	✓	✓	✓	✓	✓	Y	Y	N
Yoga Vista (V3.6.5)	D ^¤^	✓	✓	✓	✓	✕	✕	✓	✓	N	Y	N
Cleo—My MS App (V1.11.7)	A	✕	✓	✕	✕	✕	✕	✓	✕	N	Y	N
Get Steady—Balance Exercises (V1)	A ^¤^	✕	✓	✕	✕	✕	✕	✓	✕	N	Y	N
PhysioEd. (V6)	C ^¤^	✕	✓	✕	✓	✕	✕	✓	✓	Y	Y	Some

Key: ^^^ A: <50; B: 50-100; C: 101-1000; D: 1001-2000; E: 2001-5000; F: 5001-10000; G: >10000. ^¤^ Number of videos. Each video includes a number of different exercises. ^∞^ Number of games. Different mobility exercises can be set up to play a specific game depending on where the sensors are placed and the instructions of the therapist. ^β^ Difficult to determine due to difficulty with app exercise navigation. NA: not applicable.

**Table 4 healthcare-12-00929-t004:** Mobile App Rating Scale (MARS) ratings for the included apps reviewed on Apple devices (in ranked order from highest to lowest app quality mean score). Shaded apps were no longer available for download in Australia in December 2023.

App Name and Version	MARS Section Mean Score				App Subjective Quality Score (E)	App Quality Mean Score ((A + B + C + D) ÷ 4)
	Engagement (A)	Functionality (B)	Aesthetics (C)	Information (D)		
Connected mHealth (V1.3.0)	3.20	4.63	4.00	3.81	3.38	3.89
LusioMATE (V1.4.3)	3.40	3.75	4.17	3.42	3.50	3.68
Yoga Vista (V3.6.5)	3.50	4.25	3.83	2.83	2.50	3.60
Track Rehab (v3.03.3)	2.50	4.00	3.67	3.58	2.75	3.44
PhysioTherapy eXercises (PTX) (V W2.0.0)	2.20	3.75	3.00	4.43	2.75	3.34
PhysioTools Trainer (V1.0.904)	2.70	3.88	3.33	3.00	2.25	3.23
Swiss Parkinsons (V1.6.1)	2.80	3.25	3.33	3.50	2.25	3.22
Physiotec (V1.8.4)	2.30	3.75	3.33	3.40	2.38	3.20
Rehab Guru Client (V3.0.3)	2.90	3.38	3.50	3.00	2.50	3.19
PhysiApp (V2.2.2)	2.90	3.38	3.17	3.33	3.13	3.19
My TRcare—Stroke Exercises (V1.1.2)	2.60	3.50	3.00	3.50	1.50	3.15
My Exercise Program (V1.0.4)	2.10	3.75	3.00	3.50	1.38	3.09
110 Fitness (V1.4)	2.30	3.50	3.50	2.42	1.75	2.93
Daily Dose PD (V7.800.1)	2.40	3.88	2.67	2.67	2.25	2.90
Rephysio (V2.1.5)	2.10	1.63	2.83	1.83	1.00	2.10
Cleo—My MS App (V1.11.7)	3.80	4.25	4.33	4.00	3.38	4.13
Get Steady—Balance Exercises (V1)	2.00	3.13	2.33	2.67	1.00	2.53
PhysioEd. (V6)	1.90	2.63	2.83	2.67	1.13	2.50
Mean (SD) score	2.6 (0.55)	3.6 (0.67)	3.3 (0.53)	3.2 (0.62)	2.3 (0.82)	3.2 (0.49)

**Table 5 healthcare-12-00929-t005:** App Behaviour Change Scale (ABACUS) score for the included apps reviewed on Apple devices (in ranked order from ABACUS score total highest to lowest). Shaded apps were no longer available for download in Australia in December 2023.

App Name and Version	ABACUS Section Score				ABACUS Total
	Knowledge and Information (5 Items)	Goals and Planning (3 Items)	Feedback and Monitoring (7 Items)	Actions (6 Items)	
LusioMATE (V1.4.3)	2	2	5	1	10
Connected mHealth (V1.3.0)	4	0	4	1	9
PhysiApp (V2.2.2)	4	0	2	2	8
Rehab Guru Client (V3.0.3)	3	0	2	2	7
My Exercise Program (V1.0.4)	2	0	3	1	6
Swiss Parkinsons (V1.6.1)	3	0	1	2	6
Track Rehab (v3.03.3)	2	0	2	2	6
110 Fitness (V1.4)	3	0	1	1	5
Daily Dose PD (V7.800.1)	4	0	0	1	5
PhysioTherapy eXercises (PTX) (V W2.0.0)	2	0	2	1	5
PhysioTools Trainer (V1.0.904)	2	0	2	1	5
My TRcare—Stroke Exercises (V1.1.2)	2	0	1	1	4
Physiotec (V1.8.4)	2	0	1	1	4
Rephysio (V2.1.5)	2	0	0	2	4
Yoga Vista (V3.6.5)	3	0	1	0	4
Cleo—My MS App (V1.11.7)	3	0	4	2	9
Get Steady—Balance Exercises (V1)	2	0	0	2	4
PhysioEd. (V6)	1	0	0	1	2
Mean (SD) score	2.6 (0.86)	0.1 (0.47)	1.7 (1.49)	1.3 (0.59)	5.7 (2.14)

## Data Availability

All data are available in the tables.

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
