# Peer review of "Digital Apps to Improve Mobility in Adults with Neurological Conditions: A Health App-Focused Systematic Review"

_healthcare, 2024, doi:10.3390/healthcare12090929_

Round 1
Reviewer 1 Report
Comments and Suggestions for Authors
The paper presents a review of Digital Apps to Improve Mobility in Adults with Neurological
The paper is well written with detailed sections and data. The strong points of the paper are the process of conducting the review. The inclusion of different methods, parameters, and particularly experts from different domains strongly influence the quality of the review. Each section is well-detailed with necessary discussion. Relevant references are used where needed. Some weak points need to be addressed to further improve the quality of the paper.
1. The organization of the paper needs to be added at the end of the introduction section.
2. The following points listed in the abstract are the main objectives of the review.
• Which mobile health (mHealth) apps are suitable for prescribing mobility exercises for adults with neurological health conditions.
• How well these apps rate in terms of quality and capacity for behaviour change.
• How well these apps incorporate telehealth strategies.
3. Although detailed results are presented in the tables supported by supplementary data, it is difficult to answer these questions. It would be better to include a paragraph to discuss the answers considering the results presented in the table.
4. The column “Requires Companion Device (Y/N/ Optional)” in Table 1 can be removed for better readability. A single instance of the application with the yes option can be mentioned in the footnotes of the table.
5. The conclusion section needs to be in line with the findings and other sections. In the conclusion, the authors state “This review did not identify any clear recommendation for a specific app for use in the clinical setting”. This statement does not justify the earlier claims.
Author Response
Dear Reviewer 1,
Thank you for your feedback. We have responded to each of the points in the attached word document.

Reviewer 2 Report
Comments and Suggestions for Authors
-
This article presents a study summarising the use of intelligent applications using mobile and electronic technologies in the field of physical rehabilitation for people suffering from motor deficiencies and neurological disorders. Although the overall strategy of the study is relevant, it does not generally obey the essential parameters for achieving a classification such as that of mobile applications in this article. The manuscript is well prepared, but there are shortcomings in the conduct of this article and there are some issues to be resolved. Detailed comments are listed below:
- The introduction is very simplistic as it does not reflect the essence and importance of the work carried out, while the conclusion needs to be reworded to better present the results obtained. It should highlight the benefits of the work proposed. It is advisable to improve its clarity and appeal to readers.
- In addition, to enhance the relevance of the article, the authors should carry out a comparative study or at least a comprehensive bibliographical search on the subject in question. Authors should highlight the challenges and new contributions of their study, supported by appropriate and targeted references.
- Some comparative tables should be supported by diagrams and curves that better illustrate the relevance of the assertions and the results obtained by the authors.
- All results obtained should be detailed and explained.
- A more insightful commentary on the results presented in the tables and figures is required.
No Comment
Author Response
Dear Reviewer 2,
Thank you for taking the time to review this manuscript. Please find the detailed responses in the attached Word document and the corresponding revisions/corrections highlighted/in track changes in the re-submitted files.

Reviewer 3 Report
Comments and Suggestions for Authors
The paper is well structured with abstract, introduction, methodology, results, discussion and limitations. The aim of the study is well described to be identifying the gaps and challenges in existing mHealth apps for neurological rehabilitation and consisder offering practical recommendations for clinicians. Guidelines or criteria for evaluating app quality and stability would aid clinicians in making informed decisions. Adding patient perspectives in the discussion sections would help provide insights on the usability and overall satisfaction with mHealth apps. Limitations section addresses the challenges, adding more points on the impact on the overall study and future scope of work would help readers.
Author Response
Dear Reviewer 3,
Thank you very much for taking the time to review this manuscript. Please find detailed responses in the Word Document attached and the corresponding revisions/corrections highlighted/in track changes in the re-submitted files.

Reviewer 4 Report
Comments and Suggestions for Authors
In this study, research was conducted on the suitability of mobile-based applications developed in the field of health services for patients with neurological problems. Here, mostly the features of the applications for the healthcare domain are analyzed. The references given in the study are mostly related to the health sector. In this context, the study will guide researchers and physicians in this field in many ways. However, for the study to contribute to the literature in a better way, the following points should be clarified:
1) The study considers applications developed for mobile-based devices with iOS and Android operating systems. Are there software applications in this field for smart devices with other operating systems?
2) As with desktop and web-based applications, mobile-based applications need to comply with certain software development standards. For example, applications need to be scalable, testable, and maintainable. In this context, have the applications been developed with clean code architecture in mind?
3) When developing mobile-based software, the user interface of the application, memory, energy and battery management and security-related contents are important. Why is there not enough information on this subject in the study?
4) Wouldn't it be better if the references given in the study were expanded to include technical considerations in application development?
Author Response
Dear Reviewer 4,
Thank you very much for taking the time to review this manuscript. Please find detailed responses in the Word document attached and the corresponding revisions/corrections in track changes in the re-submitted files.

Round 2
Reviewer 2 Report
Comments and Suggestions for Authors
Looking at the improvements made by the authors in response to the reviewers' criticisms, I can see that the authors have taken into account most of the comments made to them. They have therefore corrected their text and enriched it with beneficial additions and changes that make their comments and the quality of their work more appreciable.